# Respiratory Manifestations in Systemic Lupus Erythematosus

**DOI:** 10.3390/ph14030276

**Published:** 2021-03-18

**Authors:** Salvatore Di Bartolomeo, Alessia Alunno, Francesco Carubbi

**Affiliations:** 1Department of Medicine, ASL 1 Avezzano-Sulmona-L’Aquila, 67039 Sulmona, Italy; salvatore.dibart@gmail.com; 2Rheumatology Unit, Department of Medicine, University of Perugia, 06123 Perugia, Italy; alessia.alunno82@gmail.com; 3Internal Medicine and Nephrology Unit, Department of Life, Health & Environmental Sciences, University of L’Aquila and Department of Medicine, ASL 1 Avezzano-Sulmona-L’Aquila, 67100 L’Aquila, Italy

**Keywords:** systemic lupus erythematosus, airway disease, interstitial lung disease, shrinking lung syndrome, diffuse alveolar hemorrhage, pleurisy, infection

## Abstract

Systemic lupus erythematosus (SLE) is a chronic systemic autoimmune disease characterized by a wide spectrum of clinical manifestations. The respiratory system can be involved in up to 50–70% of patients and be the presenting manifestation of the disease in 4–5% of cases. Every part of the respiratory part can be involved, and the severity can vary from mild self-limiting to life threatening forms. Respiratory involvement can be primary (caused by SLE itself) or secondary (e.g., infections or drug toxicity), acute or chronic. The course, treatment and prognosis vary greatly depending on the specific pattern of the disease. This review article aims at providing an overview of respiratory manifestations in SLE along with an update about therapeutic approaches including novel biologic therapies.

## 1. Introduction

Systemic lupus erythematosus (SLE) is a chronic, systemic autoimmune disease with a relapsing–remitting course and characterized by the production of a wide range of autoantibodies. Although people of any age and gender can be involved, females of childbearing age are the most affected, with a female-to-male ratio of about 9:1 [1].

SLE can have a wide range of manifestations, involving virtually every organ or apparatus, and its severity can vary from very mild disease without major organ involvement, to severe life-threatening conditions. Clinical manifestations may include cytopenia, fever, malar and other skin rashes, oral ulcers, polyarthralgia/non erosive arthritis, vasculitis, renal, neurological, cardiac and pleuro-pulmonary involvement [2,3,4]. Recently, a new set of classification criteria was proposed by American College of Rheumatology/European League Against Rheumatism (ACR/EULAR), designed to increase classification sensitivity and specificity for inclusion in SLE research studies and trials [5]. Furthermore, recommendations on disease management from EULAR were recently updated [6,7].

SLE pathogenesis is multifactorial and not completely understood, and includes an interaction between non-Mendelian genetic predisposition, hormonal and environmental factors, ultimately leading to an alteration in both innate and adaptive immunity. In particular, SLE pathogenesis is characterized by an impaired apoptotic cell clearance by phagocytes, B-cell and T-cell autoreactivity leading to an abnormal production of autoantibodies, and immune complexes (ICs) formation with nuclear and cytosolic antigens. ICs can, in turn, activate the classical pathway of the complement system contributing to inflammation and damage in target organs [4,8].

Although the exact prevalence is unknown, respiratory tract involvement can be present in 50–70% of SLE patients, being the presenting symptom of the disease in 4–5% of cases and more frequent in men [8,9,10]. Every part of the respiratory tract can be involved: upper and lower airways, vessels, pleura, lung parenchyma and respiratory muscles (Figure 1). Respiratory manifestations can be acute or chronic, primary (directly caused by the disease) or secondary (due to concomitant complications such as infections). Interestingly, acute manifestations may be associated with generalized lupus disease activity, while chronic complications may progress independently to general disease activity [10].

Respiratory manifestations of SLE are associated with a variable mortality rate, depending to the type of involvement, its extension, and the presence of comorbidities. In particular, pulmonary involvement is associated with higher mortality and with negative effect on patient-reported outcomes, patient-performed outcome and quality of life [11]. Unfortunately, clinical and therapeutic trial data specifically focused on respiratory manifestations of SLE are scarce, so treatment options are based on evidence from other organ involvement in SLE, or from respiratory manifestations in other autoimmune diseases, or based on case reports or small cases series.

In this review, we provide an overview of the scientific literature about the respiratory involvement in SLE, and highlight the progress achieved so far in the understanding of pathogenic mechanisms and in the identification of therapeutic strategies needing to be addressed in future studies. In particular, we designed a comprehensive literature search on this topic, by a review of reported published articles in indexed international journals up until 31st October 2019, following proposed guidelines for preparing a biomedical narrative review [12].

## 2. Airway Disease

Laryngeal involvement can occur in 0.3–30% of SLE patients and range from asymptomatic to severe life-threatening upper airway obstruction [13]. Clinical manifestations are non-specific and include hoarseness, cough, dyspnea, and stridor. Mucosal inflammation with erythema and edema is the major manifestation; other findings include vocal cord paralysis, bamboo nodes of the vocal cords, recurrent laryngeal neuropathy, epiglottitis, rheumatoid nodules [14], vasculitis, inflammatory mass formation and late subglottic stenosis. It usually responds well to corticosteroids (CS) therapy. However, in severe cases of respiratory failure, advanced airway management may be necessary [13,15,16].

Other airway involvement includes upper airway angioedema, necrotic tracheitis and early post-intubation stenosis, bronchial stenosis; small airway obstruction with bronchiolitis is found in the 13% to 21% of patients with the use of high-resolution computed tomography (HRCT) [17] and bronchiectasis as a consequence of direct SLE involvement or as sequelae of bronchopulmonary infections [17,18,19,20,21].

Using pulmonary function tests (PFTs), Andonopoulos et al. found a prevalence of obstructive disorders in 6% of SLE patients and 0% of control group (smokers were excluded) and initial damage of small airways (defined as maximum expiratory flow-volume (MEFV) 25–75 below 60% of predicted value) was present in 24% of SLE patients but the difference was not statistically significant with the control group [22], moreover, surveillance of pulmonary function tests revealed a progressive decline in values indicating small airways damage with time [17].

## 3. Parenchymal Lung Disease

### 3.1. Acute Diseases

Acute lupus pneumonitis (ALP) and diffuse alveolar hemorrhage (DAH) are acute and uncommon manifestations of SLE [10].

#### 3.1.1. Acute Lupus Pneumonitis

ALP is a rare, probably under-recognized, manifestation of SLE that occurs in 1–8% of SLE patients, in particular younger patients and patients with a recent diagnosis. Moreover, it can be the first manifestation of a previously unrecognized SLE in 50% of cases [10,17,23,24,25]. Clinical presentation is non-specific and can simulate infectious pneumonia with sudden onset of fever, cough, dyspnea, pleuritic chest pain and occasionally hemoptysis. Physical examination can reveal tachycardia, tachypnoea hypoxemia, hypocapnia and lung crackles. Occasionally, it can present with acute respiratory failure requiring mechanical ventilation. ALP has been described complicating SLE during pregnancy [10,17,23,24,25,26]. Chest X-ray can show multiple, bilateral patchy infiltrations, predominantly in the lower lobes, with or without pleural effusion. However, chest X-ray can be normal, especially in the initial phases or shows only lung nodules. Although these findings are non-specific, CT scan can show ground glass opacities and areas of consolidation, predominantly in the lower lobes [10,23]. Histologically, ALP presents diffuse alveolar damage (DAD) with inflammatory cell infiltration, damage and necrosis of alveolar-capillary unit, edema, hyaline membrane formation and alveolar hemorrhage. Capillaritis and thrombosis have also been described. Alveolar damage may be mediated by the deposition of ICs and complement fractions. However, there are not diagnostic and/or pathognomonic findings specific for ALP. Some data highlight a pathogenetic role of anti-Ro/SSA antibodies, due to an association between ALP and these autoantibodies [10,17,23,24,25,26]. Since there are no specific clinical or imaging findings in ALP, the diagnosis is of exclusion and a comprehensive differential diagnosis must be considered with infections, organizing pneumonia, malignancy, DAH, pulmonary edema, lung drug toxicity [23,24]. Infections must always be ruled out, since they may have a similar clinical picture and immunosuppressive treatments needed to treat ALP, could have a deleterious effect on the infection course. In this setting, bronchoscopy with bronchoalveolar lavage fluid (BALF) analysis should be performed and followed by microbiological tests for common and opportunistic pathogens [23]. It seems that the presence of eosinophilia or neutrophilia on BALF carries worse prognosis than lymphocytosis. A marked elevation in C-reactive protein (CRP) and procalcitonin levels in the serum may suggest an infection. Lung biopsy is rarely necessary [23,24,27]. Prognosis is severe, with a high mortality risk; in particular, Matthay et al. reported a mortality rate of 50% among 12 patients treated for ALP [28] while more recently Wan et al. found a mortality of 40% [29]. High doses of CS are the mainstay of treatment. In severe cases daily pulses of methylprednisolone (up to 1000 mg/day for 3 days) can be used, followed by 1–2 mg/kg per day of prednisone and a subsequent tapering according to clinical response. Immunosuppressants such as cyclophosphamide (CYC) and azathioprine, biologics drugs such as rituximab (RTX), intravenous immunoglobulins (IVIg) or plasma exchange can be added in severe refractory cases, but the evidence on their efficacy is scarce. A broad-spectrum antibiotic coverage should be started until an infection is ruled out, and then prophylaxis against opportunistic pathogens (e.g., Pneumocystis jirovecii) can be considered during immunosuppressive treatment [10,17,23,24,25,28,29]. Factors that seem to contribute to poor outcome include intercurrent infections, aspiration, diaphragmatic dysfunction, cardiac and renal failure, drug and oxygen toxicity [7,29,30,31]. Of those who recover from the acute episode, 50–100% may eventually develop chronic interstitial pneumonia so a thorough follow-up is advisable [10,31].

#### 3.1.2. Diffuse Alveolar Hemorrhage

DAH, first described by Dr. William Osler in 1904, is a rare, but very severe and potentially fatal complication of SLE [8,32]. It is not exclusive to SLE, occurring in several other conditions such as anti-neutrophil cytoplasmic autoantibody (ANCA)-associated vasculitis, antiphospholipid syndrome (APS), other connective tissue diseases, infections, bone marrow transplantation, and drug toxicity [33,34].

DAH prevalence among SLE patients ranges from 0.5–0.6% to 5.4–5.7% with a femal-to-male ratio of approximately 6:1. DAH was described as initial manifestation of SLE in 11–20% of cases; some autoptic studies in SLE patients have found the presence of red blood cells in the lungs of 30–66% of cases maybe due to the presence of either unidentified or subclinical, paucisymptomatic forms of DAH [10,33]. Mean age of presentation is 27 years, but it can occur at an early stage of the disease [17]. Some patients may have recurrent episodes [33,35].

The clinical picture of DAH is characterized by the sudden onset, within hours or a few days, of dyspnea, hypoxemia with possible acute respiratory failure and need for mechanical ventilation in more than 50% of cases, fever, cough, hemoptysis with a rapid fall in hemoglobin levels, and appearance of new alveolar or interstitial infiltrates. Some patients can present chest pain. Hemoptysis can be of variable severity, dramatic in some cases, or initially absent in up to 33% of cases [8,10,33,36].

Chest X-ray can be normal or show bilateral, rarely unilateral, airspace opacities (patchy, focal or diffuse). CT scan may show diffuse, bilateral and patchy alveolar infiltrates, also asymmetrical, ground glass opacities or diffuse nodular opacities and it is more accurate than chest X-ray to evaluate the extent of the disease. BALF is usually hemorrhagic, and the presence of 20% or more hemosiderin-laden macrophages in BALF is a criterion for DAH diagnosis [8]. However, this pattern can appear only after 48–72 h from symptom onset. BALF culture is mandatory to exclude an infection as a cause of DAH; many pathogens such as Legionella pneumophila, Strongyloides stercoralis and Cytomegalovirus can be associated with DAH [8]. Secondary infections, mainly nosocomial, can complicate the course of DAH thereby worsening the disease prognosis. Zamora et al. found a mortality rate of 100% in 3 patients with secondary infections (1 infected with Aspergillus, 1 with Escherichia coli and 1 with both methicillin-resistant Staphilococcus aureus and Candida) [37]; in a study by Rojas-Serrano et al., bronchoscopic assessment performed during the first 48 h of admission in 13 SLE patients demonstrated infections in 57% of cases including Pseudomonas aeruginosa, Serratia marcescens, Citrobacter freundii, and Aspergillus fumigates [38].

Lung biopsy is rarely necessary, and critically ill patients might not tolerate this invasive procedure. Histologic findings are non-specific with the presence of mild blood extravasation. More severe cases present capillaritis with neutrophil infiltration of alveolar septa [8,10,33,35,36]. Laboratory findings can show a rapid drop in hemoglobin levels, along with other characteristics of an active SLE, such as low complement levels, thrombocytopenia and autoantibodies. A rapid fall in hematocrit levels must alert clinicians to DAH [8]. An increase of carbon monoxide diffusing capacity (DLCO) of 30% or more over baseline values or an absolute elevation over 130% of predictive value is supportive to the diagnosis of DAH, due to the enhanced uptake of carbon monoxide by hemoglobin present in the alveoli [8,10,39].

DAH pathogenesis is not completely known, but it is characterized by an immune mediated damage of small vessels and alveolar septa, with deposition of ICs and complement fractions in the alveolar capillaries. A neutrophil interstitial infiltration with alveolar and capillary walls necrosis (capillaritis) has also been demonstrated. Neutrophils may play a pathogenetic role by the release of neutrophils extracellular traps (NETs) and cytotoxic proteins that contribute to the local damage. The loss of integrity of the alveolar-capillary wall results in the leakage of red blood cells into the alveolar space [8,10,36]. Other proposed mechanisms include: increased apoptosis of the alveolar wall cells with monocyte-macrophage infiltration, diffuse alveolar damage with edema of alveolar septa and formation of hyaline membranes, and fibrinoid necrosis. B-lymphocytes may play a pivotal role in autoantibodies formation [8,36].

Risk factors for the development of DAH include: history of thrombocytopenia, low C3 fraction, high titers of anti-double-stranded (ds)DNA, leucopenia, coexisting neuropsychiatric lupus, high disease activity (e.g., SLE Disease Activity Index (SLEDAI) score >10) and the presence of active renal disease (in particular class III and IV lupus nephritis) [8,10,36].

DAH treatment is based on case reports, expert opinion or derived from other conditions [36]. The treatment’s mainstay is the early administration of high dose iv methylprednisolone (usually 1 g/day iv for 3 or more days up to 4–8 g total dose) with subsequent tapering according to clinical evolution. CYC can be added in severe forms but data on its efficacy are contrasting with an increased mortality in the study of Zamora et al. [37], when compared to the beneficial effect in the study of Sun et al. [40]. However, a recent meta-analysis did not confirm an association with CYC and survival [41]. Other immunosuppressants have been used, such as cyclosporine, azathioprine, tacrolimus, mycophenolate mofetil (MMF), without any conclusive evidence. Among biologic drugs, RTX has shown some good results and different schemes and dosages has been used, mainly 375 mg/m^2^ weekly × 4 or fortnightly × 2 or 1 g 2 weeks apart, generally in association with CS. In the majority of reports, one course of therapy was sufficient; however, in refractory cases, maintenance therapy with RTX can be needed [8,36,42,43,44,45,46]. The potential role of belimumab remains unknown [8,36].

Plasmapheresis is generally used in patients with refractory and more severe disease, with contrasting results in literature [41]. Adverse events can occur in up to 10% of cases, are more frequent in the first procedure and are generally mild or moderate, including access site or device problems, hypotension and syncope, tingling, urticaria, nausea/vomiting, chills, fever, arrhythmia [47].

Other therapeutic options include IVIg, intrapulmonary administration of recombinant factor VIIa, and umbilical cord mesenchymal stem cell transplantation [36]. Supportive and resuscitative treatments must be guaranteed, in particular in the context of respiratory failure in which patients may require mechanical ventilation up to extracorporeal membrane oxygenation support in more severe cases. Broad spectrum antimicrobic therapy is mandatory, since infections can both initiate or complicate the course of DAH [8,36].

Prognosis is poor, with a mortality rate of up to 70–92%, (average 50%); however, a trend in the reduction of mortality was observed in the recent years, likely due to a better knowledge of the disorder, a more rapid diagnosis and a precocious introduction of novel, targeted therapies [8]. Older age, longer lupus disease duration, acute massive hemoptysis, requirement of mechanical ventilation and plasmapheresis treatment, thrombocytopenia (not universally accepted) and infections are associated with an increased risk of mortality [8,10,41]. However, severe diseases rendered the requirement of plasmapheresis treatment and mechanical ventilation are themselves associated with poor outcome. The presence of other comorbidities must also be considered. Among survivors, 70–90% can eventually develop pulmonary fibrosis therefore a strict follow-up is mandatory [10,41]. Randomized trials of therapeutics are needed to determine the most efficacious strategies for SLE-associated DAH for better management of this life-threatening complication.

### 3.2. Chronic Diseases

Chronic interstitial lung disease (ILD) in SLE seems to be less frequent in comparison to other connective tissue diseases (CTDs), and it is rarely severe [10,48,49,50]. The exact prevalence is probably underestimated, because older studies performing chest X-ray have shown the presence of ILD in 6–24% of SLE patients, while in those using a more sensitive method such as HRCT, ILD was found in up to 70% of cases, suggesting that the condition is frequently subclinical [10,49,51]. Risk factors for ILD include older age, late-onset SLE, illness duration (≥1 year), tachypnea, low levels of anti-dsDNA, high level of C3 and male gender [48,49,50,51,52]. The presence of Raynaud’s phenomenon, swollen fingers, sclerodactyly, telangiectasia, nailfold capillary abnormalities among SLE patients was associated with a higher prevalence of restrictive deficit and reduced DLCO, probably in the context of overlap syndromes that seem to carry a worse lung prognosis. Some associations were found with anti-U1 RNP, anti-SSB, anti-Scl70 and anti-SSA antibodies and sicca syndrome [10,49,50,51,52,53].

The most common pattern, histologically and radiologically, is non-specific interstitial pneumonia (NSIP); however, usual interstitial pneumonia (UIP) is not uncommon [52]. Lian et al. reported that the most frequent findings were ground glass opacities (84.4%), followed by consolidation (21.1%), honeycombing (15.6%), and traction bronchiectasis (12.8%) [53].

Clinically, ILD can evolve as a consequence a disease with acute onset (ALP or DAH) or follow a more insidious onset with chronic non-productive cough, exertional dyspnea and non-pleuritic chest pain. The mean age of onset is earlier when following an acute condition (mean 38 years) compared to the chronic form (46 years). Patients with a radiologically documented ILD can also be asymptomatic [10,51]. Inspiratory fine crackles may be heard upon physical examination, while the presence of digital clubbing is rare. Pulmonary function tests can show a restrictive pattern with reduced DLCO [10]. The severity of ILD does not correlate with SLE serologic markers [49].

Prognosis for SLE-associated ILD seems more favorable when compared to idiophatic pulmonary fibrosis or RA-associated ILD [50,52,54,55]. Toyoda et al. found a five-year survival rates of 92.9% calculated from the time ILD was diagnosed and the survival rate did not significantly differ between the patients with and without ILD [52].

Lymphocytic interstitial pneumonia (LIP) can complicate many autoimmune conditions and has been described in SLE patients in particular when associated with Sjögren’s Syndrome. LIP is characterized by the formation of lung cysts, an infiltration of the interstitium with polyclonal lymphocytes and lymphocytic alveolitis [10,49,56,57]. Prognosis is variable. Approximately 50–60% of patients respond to corticosteroids with stabilization or improvement of the disease, but in others there is progressive decline in pulmonary function and development of honeycomb lung. In general, death occurs in approximately 33 to 50% of patients within 5 years of diagnosis [56,57].

Organizing pneumonia (OP) has also been described as initial manifestation of SLE and regardless of SLE activity [10,49,58,59,60]. On HRCT, OP shows ground glass opacities, consolidations and peribronchovascular opacities. OP has also been described in rhupus syndrome [61]. CS are the treatment of choice. In the majority of cases patients recover within days of weeks after treatment introduction and radiographic findings show improvement in 50–86% of patients. Spontaneous resolution may occur. However, in a minority of cases, the disease may persist, and up to 30% may have a relapse after treatment withdrawal [62]. Several immunosuppressant agents, such as azathioprine, MMF, cyclosporin, CYC and plasmapheresis, have been used in various case reports. [58,59,60,61,62]. Finally, an association between SLE and pulmonary sarcoidosis has been described [10,63,64,65,66]. According to Rajoriya N et al., patients with sarcoidosis have an OR of 8.33 (2.71 to 19.4) for the development of SLE [64].

Placebo-controlled trials to guide the treatment of SLE-associated ILD are lacking. CS are, generally, the mainstay of treatment and patients usually show a good response. Immunosuppressants such as CYC, azathioprine, or MMF can be added in refractory more severe cases [10,23]. Among biologics, RTX can be used in some cases [67].

Treatments are generally well tolerated; with CYC, immuno- and myelosuppression, as well as IgG levels decreased can occur with subsequent infections that are generally non-life-threatening and do not necessitate stopping treatment [68,69]. In particular, in the study of Okada et al., only two sessions of CYC infusions among a total of 141 were postponed because of upper respiratory infections [69]. Interestingly, cumulative data show a higher frequency of adverse events, including hemorrhagic cystitis, premature ovarian failure, herpes zoster and cancer, with the oral administration, in comparison with pulse intravenous infusion of CYC, as found in the lupus nephritis [68,69,70]. Concerning the use of MMF in SLE-ILD, only one of ten patients with CTD-ILD had a diagnosis of SLE in the case series by Saketkoo et al. [71], while Fisher et al. included four patients with SLE-ILD in their retrospective study [72]. The most common side effects reported in these studies were diarrhea and leucopenia.

## 4. Vascular Diseases

### 4.1. Acute Reversible Hypoxemia Syndrome

First described in 1991 by Abramson [73], acute reversible hypoxemia syndrome is characterized by the acute onset of dyspnea, chest pain and hypoxemia. Pleural involvement may be present. It is frequently associated with a flare of SLE. Pulmonary imaging is generally normal, while PFTs may show reduction in vital capacity and DLCO [17,51]. Pathophysiology is not completely understood. An association between endothelium activation, with a high expression of vascular adhesion cell molecule-1 (VACM-1) and intercellular adhesion molecule-1(ICAM-1), and activated neutrophil and platelet sludging mediated by complement activation has been postulated as a pathogenic mechanism. These alterations can ultimately lead to endothelial dysfunction, vascular lumen occlusion by leukocyte aggregates and subsequent hypoxemia [17,51,73,74].

This condition rapidly responds to low doses of CS, usually insufficient to control SLE flares, when present together, so higher doses may be needed. Combination of high doses of aspirin can be useful [17,51], and most cases respond to therapy with rapid improvement of gas exchanges [9].

### 4.2. Pulmonary Embolism

SLE patients are at increased risk of developing deep vein thrombosis (DVT), occurring in up to 10% of patients [75], and pulmonary embolism (PE) with a 3-fold increased risk in comparison to general population [76]. Vein thromboembolism (VTE) represents the third most common cardiovascular (CV) event after myocardial infarction and stroke [77,78]. PE has a high mortality rate of up to 15%. Many risk factors have been investigated besides “classical” risk factors such as obesity, hyperglycemia and hyperlipidemia [77]. Moreover, You et al. found the following risk factors associated with PE: high body max index, hypoalbuminemia, positivity for anti-phospolipid antibodies (aPL), high levels of high sensitivity CRP and high doses of CS (>0.5 mg/kg/day) [78]. Finally, SLE patients with APS are at increased risk of DVT and PE. The prevalence of APS among SLE patients is about 30% [79].

APS can cause a hypercoagulable state by interacting and activating platelets, neutrophils and endothelial cells [78]. In particular, a metanalysis found that SLE patients with APS have a six times greater risk of developing PE than SLE patients without APS [79]. Moreover, patients with the positivity for lupus anticoagulant (LA) and high titers of IgG anti-cardiolipin (aCL) are at increased risk [80,81].

Clinical manifestations depend on the severity of vasculature occlusion, ranging from asymptomatic small vessels occlusion to massive PE with sudden right ventricular failure and acute circulatory collapse. Other symptoms of PE include pleuritic chest pain, dyspnea, hemoptysis, crepitations, tachypnea and tachycardia. Chronic PE can progress to secondary pulmonary arterial hypertension (PAH) due to the reduction of pulmonary vascular tree [49]. In addition to PAH, other non-thrombotic intrathoracic manifestations of APS associated with SLE are: DAH, adult respiratory distress syndrome (ARDS) and valvular heart disease (e.g., Libman-Sacks endocarditis) [49,82]. A rare, potentially fatal, manifestation of APS is the catastrophic APS (CAPS). CAPS is characterized by the diffuse occlusion of small vessels in three or more organs [81,82,83,84,85]. It generally develops in APS patients in association with a trigger such as infections, neoplasm or surgery. Respiratory failure is often present and can rapidly progress to acute respiratory distress syndrome (ARDS) [81,82,83,84,85].

Treatment of APS includes anticoagulation with the vitamin K antagonists (VKA), to maintain an international normalized ratio (INR) range of 2.0 to 3.0, for a definite period in a first provoked episode, indefinitely in recurrent episodes or in patients with a high-risk profile [81,85]. In patients with recurrent arterial or venous thrombosis, a higher INR range 3.0–4.0 or the addition on low dose aspirin should be considered. Common CV risk factors should be corrected, concurrently. In high-risk anti-phospholipid antibodies (aPL) carriers without history of thrombosis, prophylactic treatment with low dose aspirin can be adopted [81,85]. Hydroxychloroquine may reduce thrombotic risk both in APS and non-APS SLE patients due to its pleiotropic effects but evidence in this regard is still scarce [78,85]. Treatment of CAPS includes: elimination of triggers (e.g., infections), combination therapy with heparin, glucocorticoids and plasma exchange or intravenous immunoglobulins. B-cell depletion (e.g., RTX) or complement inhibition (e.g., eculizumab) can be considered in refractory cases. Supportive treatments in the intensive care unit may be necessary [81,85]. Recent systematic literature reviews and meta-analyses investigating direct oral anticoagulants have recommended against their use in these patients [86,87].

### 4.3. Pulmonary Arterial Hypertension

Pulmonary hypertension (PH) is classified into five major categories, according to its clinical characteristics and etiology and pulmonary arterial hypertension (PAH) associated with connective tissue diseases (CTDs) belongs to the first group and it is the second most frequent form after idiopathic PAH [88,89]. PAH is defined by the presence of an increase in mean pulmonary arterial pressure (mPAP) ≥ 25mmHg at rest (assessed by right heart catheterization (RHC)) with a normal pulmonary capillary wedge pressure (≤15 mmHg) and increased pulmonary vascular resistance (PVR) > 3 wood units (WU) [73]. Less frequently, SLE patients can present PH secondary to chronic pulmonary thromboembolism (group 4), mitral stenosis due to Libman-Sacks endocarditis (group 2), pulmonary veno-occlusive disease (group 1), ILD-associated PH (group 3) [88,89,90,91,92].

According to the REVEAL registry (Registry to Evaluate Early and Long-term Pulmonary Arterial Hypertension disease management), SLE patients display the second highest prevalence of PAH after systemic sclerosis (SSc) [93,94]. The real prevalence of PAH among SLE patients is unknown. Past studies have reported different results due to the method used for diagnosis (right heart catheterization (RHC) versus transthoracic echocardiography (TTE)) and the cut-off value used for the diagnosis [94]. The majority of patients are women with a mean age at PAH diagnosis of about 45 years, and with its prevalence and severity increasing with time from SLE onset. PAH can occasionally be the first manifestation of SLE. Usually, PAH tends to be moderate with systolic PAP of 40–60 mmHg and PVR between 5 and 15 WU [93,94,95]. Some possible risk factors for PAH are Raynaud’s phenomenon, active renal disease, vasculitic manifestations, pleuritis, pericardial effusion, ILD, SLEDAI ≤9, lack of rash, low erythrocyte sedimentation rate (ESR) ≤ 20 mm/h. Among immunological parameters associated with PAH: aPL, Anti-U1-RNP and anti-SSA/Ro have been described [94]. The pathogenesis of SLE-PAH is probably multifactorial and is not completely understood. Multiple factors such as genetic predisposition, environmental stimuli and immune system dysfunction could lead to an imbalance between vasoconstrictor and vasodilator mediators resulting in an increase in PVR [94,96]. aPL, anti-endothelial cells and anti-endothelin receptor antibodies, vasculitis, vasospasm, inflammation, decreased oxygen saturation, apoptosis and smooth muscle cell proliferation contribute to the development of the typical lesions of idiopathic PAH, such as plexiform lesions, smooth muscle cell hypertrophy, intimal proliferation, and collagen deposition [94,96]. Moreover, in SLE-associated PAH, there is an involvement of pulmonary veins and perivascular inflammatory infiltration [94,96,97].

Clinical presentation is non-specific, progressive and related to right ventricle dysfunction and includes dyspnea, dry cough, fatigue, weakness, exercise intolerance, angina, syncope, and hemoptysis; hoarseness due to recurrent laryngeal nerve compression, wheeze caused by large airway compression, and exercise-induced vomiting can be present in advanced cases. Symptoms are initially exercise-related, but in advanced cases occur at rest. With progression of right ventricle failure, lower limb edema, liver enlargement, abdominal distention and ascites may develop. Exceptionally, severe dilatation of pulmonary artery may complicate with its rupture or dissection leading to a cardiac tamponade. Physical findings may include: accentuated pulmonary component of the second heart sound, left parasternal lift, right ventricle third sound, murmurs indicative of tricuspid and/or pulmonary regurgitation, wheeze, and crackles; elevated jugular pressure may be present in advanced cases [88,94]. The gold standard for the diagnosis is RHC that can show some rough etiologic characterization. TTE is a non-invasive and low-cost method for the screening and follow-up of PAH patients. Other ancillary investigations may be used, such as HRCT of the lungs for the diagnosis of ILD, ventilation/perfusion scintigraphy for the assessment of chronic thromboembolism, pulmonary function tests that may show an isolated reduction of DLCO [88,94].

Early aggressive treatment aimed at normalizing PAP can improve survival. Vasodilators (e.g prostacyclin analogues), endothelin receptor antagonists (ERAs) (e.g., bosentan), phosphodiesterase 5 inhibitors (PDE-5Is) (e.g., sildenafil), guanylate cyclase stimulants (e.g., riociguat), prostacyclin IP receptor agonist (e.g., selexipag) and calcium channel blockers (CCB) (in those with a positive response to acute vasodilator testing) have shown good results. In more severe and/or refractory forms a combination with two or more different classes of drugs can be considered [49,88,94,98,99,100,101,102,103,104,105].

Side effects are in part shared by vasodilators agents. Limiting factors for CCB dose increasing are generally lower limb peripheral oedema and systemic hypotension. In the group of ERAs, ambrisentan and bosentan are associated with abnormal liver function tests (in the 0.8–3% for the former and in the 10% for the latter) with ambrisentan also associated with peripheral oedema [88,100,103]. Macitentan is not associated with liver toxicity, but a reduction in hemoglobin levels ≤8 g/dL was observed in 4.3% of patients in the study of Pulido et al. [88,106]. PDE-5Is side effects are mainly related to vasodilation such as headache, flushing and epistaxis and are mild to moderate [88]. The most frequent adverse events with riociguat were hypotension, dizziness, peripheral oedema, vomiting and anemia [105]. With beraprost the most adverse events (common with other prostanoids) are headache, flushing, jaw pain and diarrhea [88], while epoprostenol also carries the risk of a long-term intravenous catheter [88,102,104]. According to the GRIPHON study, most frequent adverse events with the use of selexipag are similar with therapies that target the prostacycline pathway (e.g., headache, diarrhea, nausea, dizziness) and are more frequent during the titration period [98].

Some studies have reported a beneficial effect of immunosuppressive therapy in SLE-associated PAH. Among immunosuppressants, CYC +/− glucocorticoids showed good response; other small studies evaluated RTX, MMF and cyclosporine. Immunosuppressants can be combined with vasoactive agents in more severe forms. Supportive treatments such as diuretics, anticoagulants and oxygen may be beneficial [88,94,107,108,109,110].

PAH affects quality of life and survival of SLE patients. Data from REVEAL registry reveal that CTD-associated PAH has a worse prognosis compared to idiopathic PAH; however, among CTDs-associated PAH, SLE patients seem to have a better prognosis, with a 1-year survival rate of 94% vs. 82% of SSc [93,94,111]. Cardiac failure and arrhythmias are the most frequent causes of death in patients with SLE-PAH [9,94].

## 5. Pleural Disease

Pleuritis is the most frequent lung manifestation in patients with SLE, occurring, often in association with pericarditis, in about 40–60% of patients during the course of the disease, although in autoptic studies up to 83% of patients can show signs of pleural involvement [10,112]. Of note, it is the only SLE manifestation of the respiratory system included in the diagnostic criteria [5]. Pleuritis, with or without pleural effusion, can be the first manifestation of SLE in the 3% and 1% of SLE patients, respectively [113,114]. Pleural involvement can be present also in overlap syndromes like rhupus syndrome [115]. The clinical picture can vary from asymptomatic, incidental findings on imaging, to pleuritic chest pain that is increased with deep inspiration, dyspnea, dry cough, fever and other systemic manifestations. Pleural effusion can be uni- or bilateral, usually mild to moderate, rarely massive. Occasionally pleuritis can be dry [10,49]. Pathogenesis of pleural effusion is thought to be due to ICs deposition on pleural surfaces. Histopathologic studies have shown the presence of a non-specific lymphoplasmacytic infiltration with rare evidence of IC-mediated vasculitis [115]. Pleuritic fluid is sterile, exudative, and yellow-tinged, but occasionally it can be turbidous or seroematic. It contains inflammatory cells such as neutrophils, but it can show a predominance of mononuclear lymphocytic cells, especially in longstanding cases. It also contains glucose levels similar to those of plasma (60–95 mg/dL), increased levels of adenosine deaminase, decreased levels of complement and ANA, in particular with titer ≥ 1:160. It has a greater pH (>7.35) and lower lactate dehydrogenase (LDH) levels (<500 IU/L or <2 times upper limit of normal for serum) than in patients with RA or tuberculosis. LE cells can be seen showing a low sensibility (about 40%) and a specificity of 80%. However, none of these characteristics are specific to SLE pleuritis [10,49,113,115,116,117]. Differential diagnosis may be difficult, since SLE patients can have pleural effusions for many reasons including infections, renal and cardiac failure, pulmonary embolism, and rarely malignancies. It is interesting to note that in SLE pleuritis CRP can be elevated also in the absence of infections [116]. Pleural biopsy can occasionally be necessary, only to rule out tuberculosis or malignancy [116]. Prognosis is usually favorable, with a good and rapid response to CS at medium dosage, although development of progressive pleural fibrosis leading to fibrothorax has been described. Non-steroidal anti-inflammatory drugs (NSAIDs) can be used for milder cases and spontaneous resolution can also occur. In more severe cases CS can be used (in patients already on steroid therapy an increase of dosages may be needed). In chronic forms, hydroxychloroquine can be used as a glucocorticoid-sparing agent. Major immunosuppressants (e.g., CYC and azathioprine) are not used, unless in the case of a concomitant systemic involvement. An association of IVIg and cyclosporine has been used in chronic, refractory pleural effusion. Chest drainage, pleurodesis and/or pleurectomy are rarely necessary in severe refractory cases [51,113,117,118,119].

## 6. Infections

SLE patients are at high risk of severe infections, by either common or opportunistic pathogens, the majority of which are lung infections, but also urinary tract, soft tissue and skin. Bacteria are the most commonly implicated agents, followed by viruses and fungi [120]. In the EuroLupus cohort, 36% of patients developed an infection and about 30% of deaths were related to infections in the five-year follow-up [121]. In addition, SLE patients have a higher incidence of respiratory failure and a high mortality rate for the ones admitted to the intensive care unit (ICU) with pneumonia as the most common cause of death. It is estimated that up to half of SLE patients develop major infections during the course of the disease [121,122,123].

Different causes accounting for this increased risk have been postulated. A genetic, non-Mendelian predisposition has been hypothesized, since the risk for severe infections seems to be increased prior to the development of SLE and a great number of genetic polymorphisms have been studied. Immunologic dysfunctions can involve both adaptive and innate immunity, in particular: complement deficiency, Ig deficiency, functional asplenia, altered cytokine production, impaired chemotaxis and phagocytosis are the major alterations thought to be involved [97,120,121,122,123,124]. SLE patients can present underlying structural alterations in the respiratory tract, such as respiratory muscle weakness, parenchymal disease, bronchiectasis, atelectasis with impaired local mucociliary clearance and defense against infections [97,120,121,122,123,124].

Immunosuppressants are well known risk factors for infections, both traditional (e.g., CYC, azathioprine) and new biologic agents (e.g., RTX and belimumab). CS are an often-underestimated cause of immunosuppression, especially when used in long term courses (>3 weeks), at relatively high dosage and in association with other immunosuppresants. On the contrary, antimalarials seem to have a protective role against infections both by allowing the reduction of CS dosage and by exerting a direct antimicrobial activity. It is also interesting to note that the risk of infections parallels disease activity [120].

Many pathogens can cause infections in SLE patients: Streptococcus pneumoniae is the most frequent cause of respiratory tract infections. Along with Salmonella, it is also associated with bacteriemia in the context of functional asplenia. Among fungal pathogens, Pneumocystis jiroveci, Criptococcus neoformans, Candida albicans, Aspergillus have been identified in SLE patients. Viral infections have been reported in particular with cytomegalovirus and varicella zoster virus, often in the context of a disseminated infection. SLE patients are also at increased risk for tuberculosis and infections with non-tuberculous mycobacteria [120,121,122,123,124,125,126]. Protozoa infections, also with rare pathogens such as Lophomonas blattarum, have been reported [127].

Diagnostic workup for infections in SLE patients may be challenging; infections can have an atypical course due to immunosuppression, moreover lung infections can simulate a lupus flare. In this context, infections must be always ruled out in a SLE patient with lung complaints and/or the appearance of a new infiltrate prior to increase the immunosuppressive therapy. Bronchoscopy with BALF analysis may be very useful for the isolation of pathogens and start of a targeted therapy [120,122]. A reduction of the immunosuppressive therapy for a short period may be necessary in severe cases during antimicrobial therapy in order to improve the immune response. 

Prevention of infections can be adopted with seasonal influenza and pneumococcal vaccination and with Pneumocystis jirovecii prophylaxis in at risk patients [128,129,130].

## 7. Miscellanea

### Shrinking Lung Syndrome

Shrinking lung syndrome (SLS) is a rare manifestation of SLE affecting less than 1% of SLE patients [131], with about 100 cases described to date [132]. Older papers reported a higher prevalence of 18–27%, while a prevalence of up to 7% has been described among patients with refractory SLE [132,133,134]. It was described for the first time in 1965 by Hoffbrand and Beck [135], and subsequently it has occasionally been described in other autoimmune diseases (e.g., systemic sclerosis, primary Sjögren’s syndrome, RA and undifferentiated arthritis) [136,137]. It is characterized by progressive exertional dyspnea, pleuritic chest pain and, less frequently, cough. It can be observed in every phase of the disease but usually it occurs in long standing disease, often as the only main organ involvement of SLE, with women more often affected than men. There is no correlation with SLE activity. Physical findings are often normal, sometimes bibasilar rales can be heard. Chest X-rays show reduced lung volumes, elevated hemidiaphragms (also monolateral) and less commonly basilar atelectasis due to poor chest expansion, pleural effusions and pleural thickening. CT scan is usually negative for parenchymal disease. Ultrasound and fluoroscopy have been proposed to study diaphragm mobility. PFTs show a restrictive pattern (reduced forced expiratory volume in the 1st second, forced vital capacity and total lung capacity) with a deterioration compared to previous tests, while carbon monoxide transfer corrected for lung volume (KCO) is normal. Echocardiography does not show any signs of PAH. No specific association was found between serologic markers and the disease, it was suggested an association with Anti-Ro/SSA. Since there are no specific diagnostic criteria, the diagnosis is one of exclusion [131,132,138,139,140,141]. The pathogenesis of this condition is not known, and several mechanisms have been proposed in recent years: micro-atelectasis with surfactant deficiency, phrenic nerve neuropathy, primary respiratory muscle myopathy, diaphragmatic fibrosis, steroid induced myopathy, pleural adhesions, and pleuritic chest pain with reduced chest expansion by an inhibitory reflex [135,140,142,143,144,145].

The majority of patients received high dose of CS, even with iv pulses, with improvement occurring in several weeks, but in some cases even in 48 h [140,141]; anecdotal data support the use of immunosuppressive agents such CYC, azathioprine, methotrexate, MMF after CS failure or as CS-sparing agents [132,139,141]. RTX has been shown to improve lung function and pain in some cases [146]. Choudhury et al. reported improvement of one patient treated with belimumab [132]. Theophylline has shown to improve diaphragmatic strength and improve PFT [147], beta-agonists could reduce diaphragmatic fatigue thanks to their positive inotropic effect [148], theophylline and beta agonists may be more efficacious if combined with CS [141]. An improvement in PFT after hematopoietic stem cells transplantation has also been described [149]. Physiotherapy could be useful to improve lung volumes and prevent impaired chest wall expansion, but it could be limited by pain [141]. Antalgic agents may be considered in the initial phase [140], while in severe respiratory weakness ICU admission and mechanical ventilation may be required [141]. Prognosis seems favorable, with a rapid improvement of symptoms, and progressive improvement, stabilization or only minor deterioration of PFTs, although full recovery is rare. Pain can persist for a long time, despite improvement in PFT. Death, due to respiratory failure is unusual [133,140,141]. In this regard, an early diagnosis and an appropriate treatment is mandatory.

## 8. Conclusions

SLE can affect any part of the respiratory tract, with various degrees of severity and at any phase of the disease course. Respiratory manifestations may display acute and/or chronic course and since most respiratory signs and symptoms are non-specific, differential diagnosis is often challenging. However, the early recognition and management of SLE-related respiratory manifestations is essential to prevent complications and the worsening of disease prognosis.

## Figures and Tables

**Figure 1 pharmaceuticals-14-00276-f001:**
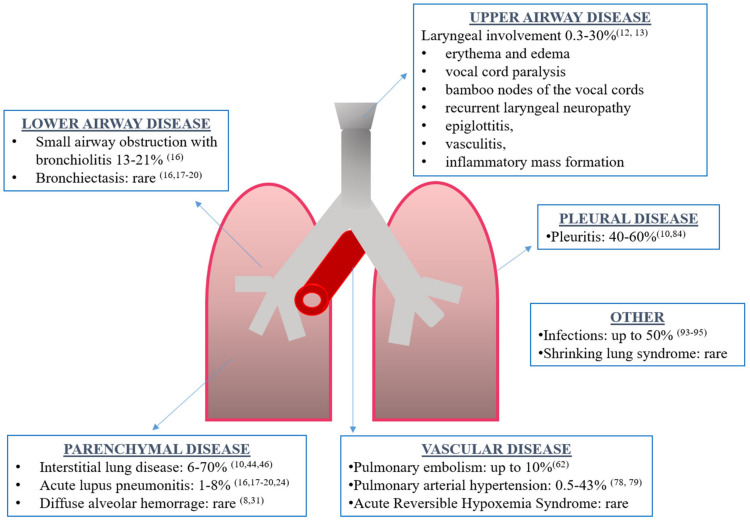
Overview of respiratory manifestations in systemic lupus erythematosus along with the prevalence and corresponding references.

## Data Availability

No new data were created or analyzed in this study. Data sharing is not applicable to this article.

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
