# Peer review of "Respiratory Manifestations in Systemic Lupus Erythematosus"

_pharmaceuticals, 2021, doi:10.3390/ph14030276_

Round 1

Reviewer 1 Report

This manuscript submission on the respiratory manifestations in systemic lupus erythematosus provides a comprehensive (with 123 references) and well written review.  It is coherent and presented in a sensible order. The review requires some attention please see the minor points below.

Minor Points

Page 8: Line 3.  Spelling demostarted – demonstrated.

Page 11. 4.1: see changes below in square brackets.

An association between endothelium ac-tivation, with a high expression of vascular adhesion cell molecule-1 (VACM-1) and intercellular adhesion molecule-1(ICAM-1), and activated [spelling: neutrophil] neuthophil [and platelet?] sludging mediated by complement activation has been postu-lated as [a] pathogenic mechanism. These alterations can ultimately lead to endothelial dysfunction, vascular lumen occlusion by leucocyte[s] aggregates and subsequent hypoxemia [16,46,60,61].  This condition rapidly responds to low doses [of] CS, usually insufficient to control SLE flares, when present together, so higher doses may be needed. Combination of high doses of aspirin can be useful [16,46].

Page 11, line 20: high doses of CS (>0.5 mg/kg/day) not (>0,5 mg/kg/die).

Page 15, line 11: ICs deposition on pleural surface, to deposition on the pleural surface or on pleural surfaces.

Page 15, line 15:  It also contains near serum level of glucose. Rewrite sentence, what are near serum levels?

Page 15, line 28: therapy incensement, should this be therapy increasement?

Page 16.  Check consistency of labelling microorganisms with lower case and upper case letters.

Page 20.  Redraft paragraph. First sentence too convoluted. Owing [to] the fact…

Final page: Check ref 118,  patient with systemic…

Final page: ref 121,  spelling of syndrome?

Final page: ref 122, Shrinking lung with syndrome?

Author Response

Author's Reply to the Review Report (Reviewer 1)

Moderate English changes required

Authors reply: We revised English language of our manuscript.

This manuscript submission on the respiratory manifestations in systemic lupus erythematosus provides a comprehensive (with 123 references) and well written review. It is coherent and presented in a sensible order.

Authors reply: We thank the reviewer for the positive comments on our manuscript.

The review requires some attention please see the minor points below. Minor Points:

Page 8: Line 3. Spelling demostarted – demonstrated.

Authors reply: We corrected this word.

Page 11. 4.1: see changes below in square brackets. An association between endothelium ac-tivation, with a high expression of vascular adhesion cell molecule-1 (VACM-1) and intercellular adhesion molecule-1(ICAM-1), and activated [spelling: neutrophil] neuthophil [and platelet? sludging mediated by complement activation has been postu-lated as [a] pathogenic mechanism. These alterations can ultimately lead to endothelial dysfunction, vascular lumen occlusion by leucocyte[s] aggregates and subsequent hypoxemia [16,46,60,61]. This condition rapidly responds to low doses [of] CS, usually insufficient to control SLE flares, when present together, so higher doses may be needed. Combination of high doses of aspirin can be useful [16,46].

Authors reply: We modified this sentence, accordingly.

Page 11, line 20: high doses of CS (>0.5 mg/kg/day) not (>0,5 mg/kg/die).

Authors reply: We modified this sentence, accordingly.

Page 15, line 11: ICs deposition on pleural surface, to deposition on the pleural surface or on pleural surfaces.

Authors reply: We modified this sentence, accordingly.

Page 15, line 15: It also contains near serum level of glucose. Rewrite sentence, what are near serum levels?

Authors reply: We modified this sentence, accordingly.

Page 15, line 28: therapy incensement, should this be therapy increasement?

Authors reply: We corrected this word.

Page 16. Check consistency of labelling microorganisms with lower case and upper case letters.

Authors reply: We thank the reviewer for the suggestion. For the binomial nomenclature of bacteria, we used upper-case characters for the first letter of the first name, and lower-case characters for the second name.

Page 20. Redraft paragraph. First sentence too convoluted. Owing [to] the fact…

Authors reply: We thank the reviewer for the suggestion . We modified this paragraph, accordingly.

Final page: Check ref 118, patient with systemic…

Authors reply: We revised this reference.

Final page: ref 121, spelling of syndrome?

Authors reply: We revised this reference.

Final page: ref 122, Shrinking lung with syndrome?

Authors reply: We revised this reference.

Reviewer 2 Report

The review summarizes respiratory manifestations involved with systemic erythematosus lupus regarding etiology, treatment, and severity. The overall contents are easy to understand and well organized.

There is a lack of detailed description of how the lung lesions of SLE affect the prognosis of SLE itself. I believe that the clinical impact of respiratory manifestations in SLE defines the significance of this review. Do most lung lesions have a little significant effect on the overall course of SLE, or do they sometimes determine prognosis?

Furthermore, it's a disappointment that the contents and structure are similar to the latest review by Amarnani et al. (Amarnani R, Yeoh SA. Denneny EK. And Wincup C., Front Med; 2021 18;7.). More attractive modifications are desired.

Author Response

Author's Reply to the Review Report (Reviewer 2)

English language and style are fine/minor spell check required

Authors reply: We revised English language of our manuscript.

The review summarizes respiratory manifestations involved with systemic erythematosus lupus regarding etiology, treatment, and severity. The overall contents are easy to understand and well organized.

Authors reply: We thank the reviewer for the positive comments on our manuscript.

There is a lack of detailed description of how the lung lesions of SLE affect the prognosis of SLE itself. I believe that the clinical impact of respiratory manifestations in SLE defines the significance of this review. Do most lung lesions have a little significant effect on the overall course of SLE, or do they sometimes determine prognosis?

Authors reply: We thank the reviewer for these suggestions. We examined this important aspect in the introduction and in the main part of our revised manuscript.

Furthermore, it's a disappointment that the contents and structure are similar to the latest review by Amarnani et al. (Amarnani R, Yeoh SA. Denneny EK. And Wincup C., Front Med; 2021 18;7.). More attractive modifications are desired.

Authors reply: We agree with the reviewer that it is indeed a disappointment that a similar review article was published in another journal just a few days before we submitted ours to this journal. However, while drafting this manuscript we simply followed a similar structure of our previous work on the topic (Alunno A et al Biomed Res Int. 2017;2017:7915340). We accept and respect the reviewer’s request of “more attractive modifications”, however since this will require a global rearrangement of the article and since the other 2 reviewers did not raise this issue, we feel appropriate to discuss this with the Editor before proceeding with these changes.

Reviewer 3 Report

This review article presents the respiratory symptoms in SLE and their treatment.

Comments:

The treatment of respiratory symptoms in SLE should be discussed in more detail. The authors should describe the potential adverse effects of this therapy.

In conclusion, it is worth mentioning the perspectives of treatment of respiratory symptoms in SLE.

Author Response

Author's Reply to the Review Report (Reviewer 3)

English language and style are fine/minor spell check required

Authors reply: We revised English language of our manuscript.

This review article presents the respiratory symptoms in SLE and their treatment.

Comments:

The treatment of respiratory symptoms in SLE should be discussed in more detail. The authors should describe the potential adverse effects of this therapy.

Authors reply: We thank the reviewer for these suggestions. We examined these important aspects of treatment in the revised manuscript.

In conclusion, it is worth mentioning the perspectives of treatment of respiratory symptoms in SLE.

Authors reply: We thank the reviewer for these suggestions. We examined this aspect in the revised manuscript.